# Obtaining New Candidate Peptides for Biological Anticancer Drugs from Enzymatic Hydrolysis of Human and Bovine Hemoglobin

**DOI:** 10.3390/ijms242015383

**Published:** 2023-10-19

**Authors:** Ahlam Outman, Mohamed Bouhrim, Codjo Hountondji, Omar M. Noman, Ali S. Alqahtani, Bernard Gressier, Naïma Nedjar, Bruno Eto

**Affiliations:** 1Laboratories TBC, Laboratory of Pharmacology, Pharmacokinetics and Clinical Pharmacy, Faculty of Pharmacy, University of Lille, 3, rue du Professeur Laguesse, B.P. 83, 59000 Lille, France; ahlam.outman.etu@univ-lille.fr (A.O.); mb@laboratoires-tbc.com (M.B.); 2UMR Transfrontalière BioEcoAgro N_1158, Institut Charles Viollette, National Research Institute for Agriculture, Food and the Environment-Université Liège, UPJV, YNCREA, Université Artois, Université Littoral Côte d’Opale, Université Lille, 59000 Lille, France; 3Laboratoire Enzymologie de l’ARN (UR6-UPMC), Université Paris Sorbonne, 75252 Paris, France; codjo.hountondji@sorbonne-universite.fr; 4Department of Pharmacognosy, College of Pharmacy, King Saudi University, P.O. Box 2457, Riyadh 11451, Saudi Arabiaalalqahtani@ksu.edu.sa (A.S.A.); 5Laboratory of Pharmacology, Pharmacokinetics, and Clinical Pharmacy, Faculty of Pharmaceutical and Biological Sciences, University of Lille, 3, rue du Professeur Laguesse, B.P. 83, 59000 Lille, France; bernard.gressier@univ-lille.fr

**Keywords:** human hemoglobin, bovine hemoglobin bovine, enzymatic hydrolysis, bioactive peptides, medicine candidate anticancer, screening

## Abstract

Enzymatic hydrolysis of bovine and human hemoglobin generates a diversity of bioactive peptides, mainly recognized for their antimicrobial properties. However, antimicrobial peptides stand out for their ability to specifically target cancer cells while preserving rapidly proliferating healthy cells. This study focuses on the production of bioactive peptides from hemoglobin and evaluates their anticancer potential using two distinct approaches. The first approach is based on the use of a rapid screening method aimed at blocking host cell protein synthesis to evaluate candidate anticancer peptides, using *Lepidium sativum* seed germination as an indicator. The results show that: (1) The degree of hydrolysis (DH) significantly influences the production of bioactive peptides. DH levels of 3 to 10% produce a considerably stronger inhibition of radicle growth than DH 0 (the native form of hemoglobin), with an intensity three to four times greater. (2) Certain peptide fractions of bovine hemoglobin have a higher activity than those of human hemoglobin. (3) The structural characteristics of peptides (random coil or alpha helix) play a crucial role in the biological effects observed. (4) The α137–141 peptide, the target of the study, was the most active of the fractions obtained from bovine hemoglobin (IC_50_ = 29 ± 1 µg/mL) and human hemoglobin (IC_50_ = 48 ± 2 µg/mL), proving to be 10 to 15 times more potent than the other hemoglobin fractions, attributed to its strong antimicrobial potential. The second approach to assessing anticancer activity is based on the preliminary in vitro analysis of hydrolysates and their peptide fractions, with a focus on the eL42 protein. This protein is of major interest due to its overexpression in all cancer cells, making it an attractive potential target for the development of anticancer molecules. With this in mind, astudy was undertaken using a method for labeling formylase (formyl-methionyl-tRNA transformylase (FMTS)) with oxidized tRNA. This approach was chosen because of the similarities in the interaction between formylase and the eL42 protein with oxidized tRNA. The results obtained not only confirmed the previous conclusions but also reinforced the hypothesis that the inhibition of protein synthesis plays a key role in the anticancer mechanism of these peptides. Indeed, the data suggest that samples containing α137–141 peptide (NKT) and total hydrolysates may have modulatory effects on the interaction between FMTS and oxidized tRNA. This observation highlights the possibility that the latter could influence molecular binding mechanisms, potentially resulting in a competitive situation where the ability of substrate tRNA to bind efficiently to ribosomal protein is compromised in their presence. Ultimately, these results suggest the feasibility of obtaining candidate peptides for biological anticancer drugs from both human and bovine hemoglobin sources. These scientific advances show new hope in the fight against cancer, which affects a large number of people around the world.

## 1. Introduction

According to the most recent data from the World Health Organization (WHO), cancer is a major cause of mortality, with an estimated 19.3 million new cases and 10 million related deaths in 2020. Forecasts for 2030 also indicate an alarming increase, with almost 26 million new cases and 17 million deaths per year [1]. The term “cancer” is commonly used to describe malignant tumors or neoplasms, which are a group of different conditions affecting various parts of the human body. These tumors result from the rapid multiplication of abnormal cells that escape the normal mechanisms regulating cell growth [2]. Current treatment options for cancer include methods such as immunotherapy, hormone therapy, stem cell transplantation, biomarker testing, and radiotherapy, with surgery and chemotherapy being the most promising options. Chemotherapy is used to disrupt this uncontrolled process of cell division [3,4]. However, many anticancer drugs are generally not specific to cancer cells, which means they can also damage healthy cells that are rapidly multiplying, leading to toxic side effects [5,6]. Furthermore, tumor cells can develop resistance to chemotherapy, limiting its effectiveness [7,8].

So, it would be advantageous to find other classes of drugs that can specifically target cancer cells without harming normal cells. These drugs should have a lower propensity to cause the development of resistance than conventional chemotherapeutic treatments. Antimicrobial peptides (AMPs) represent a promising and under-exploited alternative. AMPs can specifically target cancer cells while exhibiting reduced toxicity for rapidly proliferating healthy cells. In addition, AMPs have been shown to have a low probability of inducing the development of resistance by their target [9,10,11].

In general, bioactive peptides are oligopeptides that remain inactive within the sequence of the protein molecule but can be released through various processes such as enzymatic hydrolysis, fermentation, or gastrointestinal digestion [12,13,14]. These hydrolysates or bioactive peptides derived from proteins can be obtained from a variety of sources, whether they come from animal or plant resources, agricultural raw materials or agri-food processing, such as milk, whey, eggs, fish, marine organisms, soy, rice, peanuts, chickpeas, or even from the human body, with examples such as tears, saliva, or even blood [15,16] In this respect, hemoglobin, an agri-food protein rich in active peptides, is of particular interest due to its predominance in peptide activity databases [17]. In particular, the enzymatic hydrolysis of bovine hemoglobin generates various peptides with biological activities such as opioid [18,19], hematopoietic [20], or antihypertensive [16]. However, antimicrobial activity is the one most frequently observed and reported in scientific studies [21,22,23,24]. Human hemoglobin, like bovine hemoglobin, is an ideal substrate for proteolysis and the generation of bioactive peptides [25,26,27]. As previously mentioned, antimicrobial peptides, in addition to their role in defending against infection, have been shown to have potent toxicity towards cancer cells [2,11]. This study aimed to produce bioactive peptides from the enzymatic hydrolysis of bovine and human hemoglobin to assess their anticancer potential. The target of this study was the α137–141 peptide, known as neokyotorphin (NKT), a peptide derived from bovine and human hemoglobin, whose antimicrobial and antioxidant effects characterize it as a potential candidate for incorporation into anticancer agents [24,25,26,27].

The main aim of this study was to analyze the anticancer activity of bovine and human hemoglobin hydrolysates and their peptide fractions, using two different approaches.

The first approach consisted of using the screening process for drug candidates that inhibit protein synthesis and can be used as antibacterial and anticancer agents. This method, developed by TBC laboratories, uses the inhibition of the growth of watercress rootlets (*Lepidium sativum*) to rapidly screen drug candidates and anticancer, antibacterial, and antiparasitic drugs used or not used clinically [28,29].

In our second approach to studying the in vitro anticancer activity of peptidic hydrolysates of human and bovine hemoglobin, we initiated a study specifically targeting the eL42 protein. This protein is of major interest because it is overexpressed in all cancer cells, making it a particularly attractive potential target for anticancer molecules. Recent research has highlighted its role in the catalytic activity of the elongation step of translation, suggesting its involvement in cancer cell proliferation [30,31]. During translation in eukaryotes, the ribosomal protein eL42 binds to the CCA arm of tRNA at the 80S ribosome [31]. With this in mind, we undertook a crucial preliminary step by carrying out an experiment to label formylase (formyl-methionyl-tRNA transformylase (FMTS)) with tRNAox. This approach was chosen because of the similarities between the interaction of formylase and the eL42 protein with tRNA oxidized by its CCA arm, independently of their state (free or bound to the ribosome) [32]. The aim was to determine whether potential competition would occur resulting in the inability of the tRNA substrate to bind to the ribosomal protein in the presence of chemotherapeutic molecules. This step enabled us to gain a better understanding of how bioactive peptides could potentially interfere with tRNA and formylase, opening up new perspectives in our search for targeted anticancer molecules.

Finally, a comparative analysis of the peptide populations in the peptide hydrolysate fractions was carried out by UPLC-MS/MS.

## 2. Results

### 2.1. Effects of Bovine and Human Hemoglobin Hydrolysates and Their Peptide Fractions on the Inhibition of Protein Biosynthesis

#### 2.1.1. Study of the Influence of Initial Bovine and Human Hemoglobin Concentration on Protein Biosynthesis

Figure 1A,B show the results of the effect of the initial concentration of bovine and human hemoglobin (following a hydrolysis period of 3 h, corresponding to a degree of hydrolysis (DH) = 10%) on the inhibition of LS rootlet growth. These results were expressed in concentration–response curves. In this experimental model, it was observed that the initial hemoglobin concentration, whether bovine or human, did not influence the inhibition of LS rootlet growth. An analysis of the IC_50_ for bovine hemoglobin showed broadly equivalent values, indicating that varying the initial concentration (1%, 2%, 8%, or 10%) did not make a significant difference in its impact on the inhibition of LS rootlet growth. Similarly, increasing the initial concentration of human hemoglobin did not appear to influence the inhibition of LS rootlet growth, with very similar IC_50_. A comparison of the IC_50_ of the two species did not reveal any significant differences either (Table 1).

#### 2.1.2. Study of the Inhibition of Rootlet Growth Activity by Bovine and Human Hemoglobin Hydrolysates According to Their Degree of Hydrolysis

Studies carried out on the hydrolysis of human hemoglobin revealed that the enzymatic process involved the same reaction mechanism as that observed in bovine hemoglobin, generally referred to as the “zipper” mechanism. This mechanism generates a considerable diversity of peptides [27]. It is also important to note that different sets of peptides are produced at different degrees of hydrolysis (DH) [24]. Consequently, for the remainder of the study, it was essential to select several hydrolysates with varied peptide populations. Eight hydrolysates were chosen, with DHs of 0, 3, 4, 5, 6, 8, 10, and 18%, corresponding to hydrolysis times of 0, 5, 15, and 30 min, as well as 1, 2, 3, and 24 h, respectively.

With this in mind, the effect of the influence of the degree of hydrolysis (DH) of bovine and human hemoglobin on the inhibition of LS rootlet growth was analyzed and presented in the form of dose–response curves in Figure 2A–D. The IC_50_ values obtained are recorded in Table 2. These results show that for some DH, bovine and human hemoglobin hydrolysates showed a more marked effect in inhibiting rootlet growth than for others. For example, when examining the degree of hydrolysis (DH) of 0, the IC_50_ values were found to be 5.06 ± 2.00 mg/mL for bovine hemoglobin and 3.97 ± 0.60 mg/mL for human hemoglobin. In these circumstances, the inhibitory effects were less marked for both species compared with the other DHs. This was followed by DH of 3, with IC_50_ of 2.42 ± 0.96 mg/mL for bovine and 2.68 ± 0.67 mg/mL for human, and DH of 18, with IC_50_ of 2.25 ± 0.84 mg/mL for bovine and 2.25 ± 0.84 mg/mL for human. It should be emphasized that these results, although still significantly better than those observed for nonhydrolyzed hemoglobin at a DH of 0, demonstrated improvements of up to twice the efficacy. The other DHs showed very interesting inhibitory effects on rootlet growth, three to four times more intense than for the DH of 0. For example, DHs of 4, 5, 6, 8, and particularly 10% showed significantly greater inhibitory effects.

#### 2.1.3. Study of the Inhibition of Rootlet Growth Activity of Bovine and Human Hemoglobin Hydrolysate Fractions and Peptidomics Approach

In light of previous results that revealed marked inhibitory effects on rootlet growth, particularly at a degree of hydrolysis (DH) of 10% (equivalent to a hydrolysis time of 3 h), the fractionation of these hydrolysates was also carried out after a 3h peptic hydrolysis. The first fraction (Fraction 1) corresponded to the NKT peptide (α137–141, neokyotorphin), followed by fractions collected at 5min intervals (Figure 3A). Subsequent analyses were performed to assess their inhibitory effects on rootlet germination.

The results of this study made it possible to identify the specific bovine (B) and human (H) hemoglobin fractions with the strongest inhibitory effects on rootlet growth (Figure 3B–E). Analysis of the IC_50_ values revealed several fractions that stood out for their robust inhibitory activity (Table 3). By comparing the performance of fractions from the two types of hemoglobin, similarities as well as nuances in their bioactive potential were highlighted. Of the fractions analyzed, Fraction 1 stood out for its remarkable inhibitory efficacy. The IC_50_ values for this fraction were 29 ± 1 µg/mL for bovine hemoglobin and 45 ± 2 µg/mL for human hemoglobin. These results indicated a significant inhibitory effect for both species. In the case of Fraction 2, there was a marked contrast between the two types of hemoglobin. The IC_50_ values of Fraction 2 from bovine hemoglobin were approximately seven toeight times better than those obtained from human hemoglobin. This observation clearly suggested the superiority of bovine hemoglobin for this specific fraction. Fractions 3, 4, 5, 6, and 9 stood out for their significant inhibitory capacity towards rootlet growth. At the same time, Fraction 8, whether of bovine or human origin, also had a lesser inhibitory effect than the fractions mentioned above. What linked themwas that, despite their distinct peptide profiles from the two hemoglobin sources, they demonstrated an ability to inhibit LS rootlet growth. This uniformity in inhibitory performance between the two types of hemoglobin suggested a degree of convergence in their bioactive properties. Fraction 7 was also found to have inhibitory activity for both species, although this fraction exhibited relatively less inhibitory activity than the other fractions.

Next, triple RP-HPLC-MS/MS analyses were carried out following a 3h peptide hydrolysis on the two types of hemoglobin. A comparative analysis of the peptide populations in the fractions was carried out. To illustrate the results, a histogram (Figure 3F) showing the number of unique peptide sequences identified was used to allow precise visualization of the disparities and similarities between the peptide populations in the fractions. By cross-referencing the results, it was found that the fractions with LS rootlet inhibitory properties in both species, particularly Fractions 3, 4, 5, 6, and 7, had the highest number of peptide sequences identified. An exception was seen in Fraction 1, corresponding to the NKT peptide (α137–141, neokyotorphin, which stood out as a unique and pure fraction.

#### 2.1.4. Study of the Inhibition of Rootlet Growth Activity of Pure Antimicrobial Peptides α137–141 and α1–32 with Different Structural Characteristics

To further our investigation, this section of the study focused on analyzing the inhibitory effects on LS rootlet growth of standard α137–141 and α1–32 antimicrobial peptides (Figure 4). These two peptides had distinct structural characteristics. The NKT peptide, or α137–141, was characterized by a “random-coil” structure and a restricted number of amino acids (5), with a monoisotopic molecular weight of 654 Da. In contrast, the α1–32 peptide had an alpha helix structure (monoisotopic molecular weight: 3327 Da) and a higher number of amino acids (32) [33,34]. These antimicrobial peptides differed in their IC_50_ values, reflecting the variety of their inhibitory effects. The standard peptide α137–141 showed an IC_50_ of 53 µg/mL, indicating strong inhibitory activity. This value was consistent with the IC_50_ obtained for NKT peptides derived from bovine (29 ± 1 µg/mL) and human hemoglobin (45 ± 2 µg/mL), respectively. In contrast, the α1–32 peptide exhibited less inhibitory activity with an IC_50_ of 280µg/mL, representing a 5-fold lower value than NKT (Table 4). However, its inhibitory activity on LS rootlet growth remained positive.

### 2.2. Study of the Inhibition of Rootlet Growth Activity of Bovine and Human Hemoglobinhydrolysates and Their Fractions: Covalent Labeling of Proteins Using tRNAox

#### 2.2.1. Covalent Labeling of Formylase by tRNAox

Affinity labeling with tRNAox was developed to identify amino acid residues located at the binding site of the CCA arm of tRNA on enzymes purified from the translational apparatus. These enzymes included the aminoacyl-tRNA synthetase family, which esterified amino acids at the 3′-OH end of tRNAs, and formyl-methionyl-tRNA transformylase (FMTS or formylase). These studies led to the discovery of the consensus sequence 332Lys-Met-Ser-Lys-Ser336 as the catalytic signature of all class 1 aminoacyl-tRNA synthetases. The Lys-335 residue of this tRNAox-labeled motif was shown to be a catalytic residue that stabilized the transition state preceding the formation of the aminoacyl-tRNA product [35]. To develop tRNAox labeling of the human ribosomal eL42 protein, we reproduced a tRNAox labeling experiment for formylase (FMTS). Since any molecule that binds FMTs by preventing the FMTs-tRNAox complex interacts with the eL42 protein of the human 60S ribosome (a protein found overexpressed in a large number of human cancers), it is a good candidate for antitumor molecule research. Figure 5 shows formyl-methionyl-tRNA transformylase (FMTS or formylase) in the presence and absence of tRNAox and 4-phenyl-3-thiosemicarbasone piperitone (UCK-36) analyzed by electrophoresis on a 10% polyacrylamide gel under denaturing conditions (SDS-PAGE), revealed by Coomassie blue staining. In well 1, we deposited free FMTS; in well 2, FMTS and tRNAox; in well 3, FMTS and tRNAox in the presence of 4-phenyl-3-thiosemicarbasone piperitone (UCK-36), an inhibitor molecule that is already known to make a covalent bond with the lysine residue concerned instead of tRNAox.

In the presence of tRNAox (well 2), the 32 kDa control band corresponding to free FMTS (well 1) was transformed into a 57 kDa band corresponding to the covalent labeling of an FMTS molecule (32 kDa) by a tRNAox molecule (25 kDa) (well 2). Well 3 showed that this molecule masked the side chain of the lysine residue to prevent it from being labeled by tRNAox. It was therefore more competitive than tRNAox.

#### 2.2.2. Testing of Bovine and Human Hemoglobin Peptide Hydrolysates and Their Peptide Fractions

Figure 6 shows the characterization of formylase (FMTS) in the presence and absence of tRNAox as well as bovine (6A) and human hemoglobin (6B) hydrolysates and their fractions, performed by 10% polyacrylamide gel electrophoresis under denaturing conditions (SDS-PAGE), followed by Coomassie blue staining. The samples were arranged as follows: well 2 contained free FMTS, well 3 contained FMTS in the presence of tRNAox, and wells 4 to 8 contained FMTS bound to tRNAox in the presence of the products to be evaluated. Observation of the results revealed that, in well 2 (Figure 6A), a protein band of approximately 32 kDa was present, indicating the presence of formylase. Bands at around 32 kDa (corresponding to free FMTS) and at around 57 kDa (resulting from the covalent binding of a 32 kDa FMTS molecule to a 25 kDatRNAox molecule) were visible in wells 3, 5, 6, and 7. However, in wells 4 and 8, the 57 kDa band was less intense. This observation was made for samples containing Fraction 1 (representing bovine NKT) in well 4, as well as for the total hydrolysate in well 8. These results suggest that these samples could influence the interaction between FMTS and tRNAox by modulating the formation of the covalent bond. Similar observations were made for human hemoglobin (Figure 6B). In well 4 (containing fraction 1 representing human NKT) and well 8 (total human hydrolysate), the 57 kDa band was also attenuated. This suggests that these samples may also affect the interaction between HTSF and tRNAox. The results indicate that Fraction 1 and total hydrolysate of human and bovine hemoglobin may modulate the interaction between FMTS and tRNAox. This observation suggests that these peptide fractions could influence molecular binding mechanisms, potentially hindering efficient binding of the tRNA substrate to the ribosomal protein in the presence of chemotherapeutic compounds. In short, this experimental phase enriches our understanding of the competitive interactions between bioactive peptides and tRNA, as well as their possible interference with formylase.

## 3. Discussion

In the search for new therapeutic approaches in the fight against cancer, antimicrobial peptides (AMPs) are emerging as a promising and underexploited avenue. Possessing the ability to selectively target cancer cells while minimizing their toxicity to rapidly proliferating healthy cells, AMPs also exhibit a low tendency to elicit resistance in their target [9,10,11]. In particular, the enzymatic hydrolysis of bovine hemoglobin generates a diverse range of peptides with numerous biological activities. However, antimicrobial activity remains the most frequently documented and studied in the scientific literature [22,23,24,33]. Human hemoglobin, like bovine hemoglobin, represents an optimal substrate for proteolysis and the production of bioactive peptides [25,26,27]. With this in mind, the main aim of this study was to generate bioactive peptides using enzymatic hydrolysis of bovine and human hemoglobin, followed by analysis of their potential anticancer properties using two distinct approaches. The first innovative approach has paved the way for a promising methodology for assessing the anticancer activity of various molecules, including our bovine and human hemoglobin hydrolysates and their peptide fractions.

Initial investigations focused on the influence of initial concentrations of bovine and human hemoglobin. Contrary to expectations, no significant differences were observed between the two species. The results showed that the inhibition of LS rootlet growth was not significantly influenced by the initial concentration of hemoglobin of any kind. This finding suggests that antimitotic properties are not directly correlated with initial hemoglobin concentration. Continuing the analysis, the effect of the degree of hydrolysis (DH) on antimitotic activity was explored. The results showed that bovine and human hemoglobin hydrolysates exhibited more pronounced inhibitory effects at certain DHs than at others. In particular, DHs of 3, 4, 5, 6, 8, and especially 10% showed significantly greater inhibitory effects than those observed at a DH of 0, and these effects were three to four times more intense. This suggests that the hydrolysis process generates bioactive compounds that are favorable for inhibiting rootlet growth, and it should be noted that different sets of peptides are generated depending on the degree of hydrolysis (DH) applied. Overall, advancing the degree of hydrolysis led to an increase in the number of peptide sequences identified, particularly for peptides with masses between 0 and 1000 Da [24]. A comparative analysis of the IC_50_ fractions of the two types of hemoglobin was also carried out. This revealed that, despite the distinct peptide profiles from the two hemoglobin sources, Fractions 3, 4, 5, 6, and 9 showed comparable inhibitory effects between cattle and humans. However, for Fraction 2, the IC_50_ values obtained from bovine hemoglobin were significantly better (around seven to eight times) than those obtained from human hemoglobin. This suggests the superiority of bovine hemoglobin for this specific fraction.

Regarding our peptide of interest, peptide α137–141, NKT, identified in Fraction 1, it stood out for its particularly robust inhibition, as evidenced by significantly low IC_50_ values, demonstrating a significant capacity to restrict rootlet growth. The IC_50_ values for bovine hemoglobin were only 29 ± 1 µg/mL and for human hemoglobin were48 ± 2 µg/mL, which was 10 to 15 times lower than other fractions. This observation could be attributed to the purity of this fraction, combined with the presence of a peptide with exceptionally potent antimicrobial activity [33,36,37,38]. Furthermore, results obtained with the synthesized NKT confirmed this trend, showing particularly interesting IC_50_ values, specifically 53 µg/mL.

It is also crucial to highlight another relevant criterion. These results underscore the diversity of inhibitory effects exhibited by bioactive peptides and emphasize the significant role of their specific structural characteristics in regulating rootlet growth.

More precisely, peptide α137–141 (NKT) is characterized by a “random coil” structure and a molecular weight of 654 Da, with an exceptionally low IC_50_ of only 0.053 mg/mL (53 µg/mL). A comparison of this peptide to another synthetic antimicrobial peptide was conducted. This was peptide α1–32, with an alpha-helix structure and a molecular weight of 3327 Da, also derived from the enzymatic hydrolysis of bovine [33] and human hemoglobin [39]. This α1–32 fragment exhibited notable antimicrobial activity against a variety of bacterial strains and was identified in Fraction 6, eluting between 25 and 30 min [33,34]. This fragment showed an IC_50_ of 0.28 mg/mL (280 µg/mL), which was ten times higher than that of peptide α137–141. However, it retained significant inhibitory capacity.

These findings aligned with the antimicrobial activity observed for both peptides. Peptide α137–141 stood out for its more potent antimicrobial activity than fragment α1–32, despite its shorter amino acid sequence. Studies have even revealed that peptide α137–141 exhibits minimum inhibitory concentrations (MICs) up to 54 times lower against a specific strain, as illustrated in Table 5, demonstrating its superior efficacy in combating pathogenic microorganisms [36,37].

These observations highlight the importance of several parameters in the hydrolysis process and its impact on anticancer activity. They also underline the fact that hydrolysates and their fractions, in particular the NKT fraction, specifically target the host cell’s protein synthesis mechanism, a crucial aspect when it comes to curbing tumor proliferation. Recent advances in cancer therapy underline the importance of targeting multiple translation elements and signaling pathways. For example, the synergistic approach of targeting both HSP 90/70 and the 26S proteasome is emerging as a promising strategy for treating cancer, highlighting the crucial role of multifactorial approaches [40]. Various compounds have been identified as potential cancer therapies, targeting different translation factors [41]. For example, some antibiotics such as puromycin and sparsomycin act by mimicking aminoacyl-tRNA or blocking peptide synthetase to inhibit protein synthesis [42]. Cycloheximide is another example of a protein synthesis inhibitor, disrupting peptidyl transferase in eukaryotic ribosomes [43]. Doxorubicin (DOX) inhibits eukaryotic ribosome formation [44], while erythromycin (ERY) targets protein synthesis in prokaryotic and eukaryotic cells [45]. Doxycycline (DOC) directly disrupts protease-activated receptor 1 to slow tumor progression [46]. Monensin (MON) targets the EGFR signaling pathway, inhibiting cell proliferation and growth in chemotherapy-resistant pancreatic cancer cells [47]. Amikacin (AMK) binds to the A site of 16S RNA similarly to kanamycin A, with specific interactions between the L-(−)-γ-amino-α-hydroxybutyryl group and RNA [48]. What is particularly remarkable is that the combination of these various compounds exhibits notable anticancer potential by inducing significant inhibition of rootlet growth [28].

Subsequently, a striking aspect emerged from the results of the second method of assessing anticancer activity, namely the preliminary study centered on the in vitro analysis of the anticancer activity of hydrolysates and their peptide fractions, focusing on the eL42 protein. These findings not only reinforced the findings of the first approach but also supported the essential hypothesis that the inhibition of protein synthesis is central to the development of carcinoma. Indeed, the results suggest that samples containing the 1(NKT) fraction and the total hydrolysate for both types of hemoglobin may exert modulatory effects on the interaction between FMTS and tRNAox. This finding highlights the possibility that these peptide fractions may influence molecular binding mechanisms. In other words, there could be potential competition leading to the inability of the tRNA substrate to bind efficiently to the ribosomal protein in their presence. This observation was particularly striking for NKT, whether derived from human or bovine hemoglobin, as it was perfectly consistent with its remarkable efficacy in inhibiting LS seed germination.

These observations suggest that NKT could be one of the most promising pathways for inhibitory activity. In addition, its significant antimicrobial and antioxidant properties could qualify it as a potential candidate for inclusion in the class of anticancer agents.

In short, this experimental phase will enable us to gain a better understanding of the possible competitive interactions between bioactive peptides and tRNA, as well as their possible interference with formylase. This approach opens up new research prospects for the targeted discovery of anticancer molecules. However, further investigations are crucial to detail the mechanisms underlying these inhibitory effects and to better understand the influence of our bioactive peptides on potential competition with transfer RNA (tRNA) for binding to the eL42 protein within the human 80S ribosome.

## 4. Materials and Methods

### 4.1. Materials: Reagents, Solvents, and Standards Used

All chemicals and solvents were of analytical grade from commercial suppliers: Sigma-Aldrich (Saint-Quentin Fallavier, France) or Flandres Chimie (Villeneuve d’Ascq, Croix, France). Ultrapure water was prepared using a Milli-Q system in the laboratory. Purified bovine hemoglobin (H2625) powder, dark brown and purified human hemoglobin (H7379), and dark red were purchased from Sigma-Aldrich (Saint-Quentin Fallavier, France). The hemoglobin wasstored at 4°C before use. Neokyotorphin (NKT) standard α137–141 and Alpha 1–32 were supplied by Genecust (Luxembourg) and stored at −20°C until use. Pepsin is a lyophilized powder derived from porcine gastric mucosa and was purchased from Sigma-Aldrich (P6887, (Saint-Quentin Fallavier, France)). Pepsin activity was measured at 3350 AU/mg protein according to a protocol established by the supplier Sigma-Aldrich. Pepsin was stored at −20 °C. Watercress seed (Lepidium sativum) brand Truffaut (zone commercial Cora, Rue du 19 Mars 1962-02,100 Saint-Quentin, France).

### 4.2. Preparation of Bovine and Human Hemoglobin Hydrolysates

Conventional enzymatic hydrolysis using pepsin has been used to compare two types of purified hemoglobin: bovine hemoglobin and purified human hemoglobin. Enzymatic hydrolysis of bovine and human hemoglobin has been shown in several studies to produce active peptides, including peptide α137–141, which has a variety of biological activities [24,27,33,36,49].

### 4.3. Preparation of the Stock Solution

Stock solutions were prepared by adding 15 g of bovine (BH) or human (HH) hemoglobin to 100 mL of ultrapure water. After centrifugation at 4000 min −1 for 30 min (Eppendorf AG, Hamburg, Germany; Centrifuge 5804 R, Brinkmann Instruments, Westbury, NY, USA), the supernatant was collected. The Drabkin method, established by Crosby, Munn, and Furth in 1954, was used to determine the actual concentrations of BH (C_BH_) and HH (C_HH_) using a spectrophotometric approach to quantify hemoglobin. To do this, 20 µL of the sample was mixed with 10 mL of Drabkin D5941 reagent (Sigma-Aldrich), followed by a 15 min incubation at room temperature protected from light. Absorbance was then measured at 540 nm using a UV spectrophotometer (ChemStation UV spectrophotometer, VIS 8453A, Agilent Technologies, Santa Clara, CA, USA). The results were adjusted according to the calibration curve. From the C concentrations of the bovine or human stock solution, various solutions of hemoglobin were obtained by dilution, reaching the following specific concentrations: 1%, 2%, 8%, and 10% (*w*/*v*).

### 4.4. Hydrolysis Process

To denature the hemoglobin solution, which was initially in its native “globular” form, the pH was adjusted to 3.5 using 2 M hydrochloric acid, which was added gradually. The hydrolysis reaction was started by the addition of pepsin (EC 3.4.23.1, 3200–4500 units mg^−1^ of protein) that had previously been solubilized in ultrapure water, with an enzyme/substrate ratio equal to 1/11 (mole/mole). Samples were taken at T_0,_ T_2.5_, T_30_, T_60_, T_120_, and T_180_ min of hydrolysis, corresponding to different degrees of hydrolysis. The peptic hydrolysis reaction was then stopped by adding sodium chloride NaCl 5M to a final pH of 9, which deactivated the enzyme. The temperature was maintained at a constant 30°C throughout the reaction. The samples were stored at −20 °C, then lyophilized, and the powder recovered was ready for testing.

### 4.5. Fractionation of Peptide Hydrolysates by Semipreparative HPLC

In order to complete the research results, the total hydrolysates were fractionated. To do this, it was necessary to increase the concentration of the initial substrate used in order to recover more active peptides in a single step so that they could be used for cancer treatment. Consequently, hydrolysates could be chosen, more specifically those with a DH of 10% and hydrolysis time of 3 h. It has been shown that increasing the peptide concentration of the 10% (*w*/*v*) hydrolysate allows the recovery of up to a 10-fold higher concentration of active α137–141 peptide compared with the 10% (*w*/*v*) hydrolysate [27]. This results in a 10-fold enrichment compared with the initial hydrolysate and could suggest an interesting use for co-products. The fractions were collected every 5 min in a tube using a liquid chromatography system. This system consisted of a Waters 600E automated gradient controller pump module, a Waters Wisp 717 automated sampling device, and a Waters 996 photodiode array detector. Empower 3 software (Version 3 Waters) was used to plot, acquire, and analyze the chromatographic data. All chromatographic procedures were carried out using a semipreparative C4 column (250 mm × 4.6 mm, 3 mm internal diameter). The mobile phases were ultrapure water/trifluoroacetic acid (1000:1, *v*/*v*) as solvent A, and acetonitrile/trifluoroacetic acid (1000:1, *v*/*v*) as solvent B. Samples were filtered at 0.20 μm and then injected. Online UV absorbance scans were performed between 200 and 390 nm at a rate of one spectrum per second with a resolution of 1.2 nm [23,50]. The injection volume was 60 µL. The flow rate was 0.6 mL min −1. The elution program was as follows: The mobile phases were LC-MS grade water with 0.1% trifluoroacetic acid (1000:1, *v*/*v*) as solvent A, and LC-MS grade acetonitrile, ACN, with 0.1% trifluoroacetic acid (1000:1, *v*/*v*) as solvent B. A gradient was applied with solvent B increasing from 5% to 30% over 30 min, then to 60% for 10 min, and maintained until 47min at 95%, then back to initial conditions. The tubes containing the different fractions were dried with speedVac and stored at −20 °C.

### 4.6. RP-UPLC Analysis and Mass Spectrometry

After centrifugation for 10 min at 8000× *g*, 10 µL samples of bovine and human hemoglobin hydrolysate at a concentration of 30 mg-mL^−1^ were subjected to triplicate RP-HPLC-MS/MS analysis using an ACQUITY UPLC system (Waters Corporation, Guyancourt, France). The peptides were separated on a C18 column (150 × 3.0 mm, 2.6 µm, Uptisphere CS Evolution, Interchim, France). The mobile phases consisted of solvent A (0.1% (*v*/*v*) formic acid/99.9% (*v*/*v*) water) and solvent B (0.1% (*v*/*v*) formic acid/99.9% (*v*/*v*) acetonitrile (ACN)). The ACN gradient (flow rate 0.5 mL-min^−1^) was as follows: from 5% to 30% solvent B in 40 min, from 30% to 100% solvent B in 10 min, followed by washes and equilibrations using 100% and 1% solvent B, respectively, for 5 min each. The eluate was introduced into the electrospray ionization source of the qTOFSynapt G2-Si™ (Waters Corporation, Manchester, UK), previously calibrated with sodium solution. Mass spectrometry (MS) measurements were performed in sensitivity, positive ion, and data-dependent analysis (DDA) mode using Mass Lynx 4.2 software (Waters). The source temperature was maintained at 150°C, and the capillary and cone voltages were set at 3000 and 60 V-MS. Data were collected for m/z values between 50 and 2000 Da, with a scan time of 0.2 s. Up to 10 precursor ions were selected for MS/MS analysis, with an intensity threshold of 10,000. MS/MS data were collected in collision-induced dissociation (CID) fragmentation mode, with a scan time of 0.1 s and specified voltages of 8–9 V and 40–90 V for lower and higher molecular weight ions, respectively.

### 4.7. Determination of the Anticancer Activity of Hydrolysates and Peptide Fractions

#### 4.7.1. Lepidium Sativum (LS) Rootlet Growth Test

##### Description of the Methods Used

The anticancer efficacy of peptic hydrolysates of human and bovine hemoglobin and their peptide fractions was determined using a patented method developed by our laboratory [28]. This method is used by pharmaceutical laboratories TBC (Paris, France) for the rapid screening of candidate molecules for anticancer, antibiotic, and antiparasitic drugs. The technique involves assessing the effect of the various products tested (human and bovine hemoglobin hydrolysates and fractions) on rootlet growth during germination by measuring length. The use of LS seeds for the screening of anticancer drug candidates appears to be a promising and ethically responsible method, offering an alternative to animal experimentation. Although this method cannot completely replace studies using animal models, it represents an important opportunity to reduce the number of animals used and cut preclinical research costs. Other alternatives to animal experimentation exist, such as sophisticated tests using cell cultures and human tissues (in vitro), advanced computer models (in silico), and studies involving human volunteers. In addition, the emergence of “organs on a chip” (that simulate the structure and function of human organs) offers new prospects for preclinical studies. However, all these experimental approaches remain complex to put into practice and are associated with high costs. Nevertheless, validating the efficacy of this model is not limited to compounds traditionally used in oncology, but also extends to antibiotics and antiparasitics (all molecules or xenobiotics that inhibit protein synthesis [28]).

##### Preparation of Vegetable Seeds

LS seeds were pregerminated from seeds in Petri dishes on filter paper or blotting paper soaked in distilled water in total darkness for 24 h. The aim of pregerminating the seeds was to eliminate those that did not show signs of germination with a closed cuticle. Seeds that did not show these signs were not likely to germinate and were therefore removed before the tests begin. This ensured that only seeds suitable for germination were used in subsequent tests. The seeds were germinated in the presence of different concentrations of the products to be tested, then the results were compared toa negative control. This control consisted of normal germination under the same conditions as the treated seeds but with distilled water and no added product.

##### Preparation of Test Concentrations

Different concentrations of the products to be tested were prepared, ranging from low to high concentrations, generally from picograms (pg/mL) to milligrams (mg/mL). Next, 5 mL of each product at different concentrations was added to Petri dishes containing watercress seeds. After incubation in the dark for 3 days (72 h), the length of the rootlets was measured with a ruler and recorded in centimeters. To obtain more accurate results, it was recommended that more than 8 measurements were taken per concentration of test or control product to allow a proper comparison of the effects of each product on rootlet growth.

#### 4.7.2. Covalent Labeling of Proteins with tRNAox by Formylase

##### Preparation of tRNAox

In an Eppendorf tube, 4.8 µL of tRNA (419 µM) was taken and 4 µL of NaIO4 (0.5 M) and 2 µL of NH4OAc (5 M) at pH 5.0 were added. This mixture was made up to 50 µL with water (DNase/RNAase free) and incubated for 1 hat 4°C. After removing the NaIO4 via a G-25 column, the sample was added to the center of the resin in the G-25 column. Centrifuging at 3000 rpm for 2 min recovered a sample of approximately 110 µL. Precipitation was achieved by adding 0.2 times the sample volume of NaCl (5 M) to the tRNAox, i.e., 22 µL of NaCl and 2.2 times the sample volume of cold ethanol −20 °C, i.e.,242 µL of ethanol. The mixture was then prepared by adding −20 °C cold ethanol to 700 µL, homogenizing, and leaving it to stand overnight at −20 °C. Finally, by centrifugation at 13,000 rpm at 4°C for 45 min, the already oxidized tRNA was concentrated and the ethanol was allowed to evaporate. The precipitate obtained was resuspended in 1 mL of water (without DNase/RNAase), giving tRNAox prepared at 2000 pmoles. The tRNA oxidation and protein labelling reactions are shown in Figure 7:

The principle of covalent labeling involves the use of a reactive analog of tRNA as an affinity indicator targeted to the catalytic site of purified enzymes within the translation machinery, as well as to the A, P, and E sites of ribosomes. This analog is oxidized tRNA (tRNAox), which is tRNA with the 2′, 3′ cis-diol group of the 3′-terminal adenosine ribose converted to the 2′, 3′ dialdehyde group by sodium periodate [31]. The aldehyde function in the 2′ or 3′ position of the tRNAox can form a Schiff base with an amine function of a lysine (or arginine) residue of a protein. This Schiff base is then stabilized by a specific reducing agent, sodium cyanoborohydride (NaBH3CN), leading to the creation of a covalent complex between the tRNA and the lysine residue of the protein [31] (see Figure 8). Labeling of formylase using tRNAox outside the ribosome was performed by adding nonradioactive tRNAox at a concentration of 100 µM to a 10 µL reaction mixture (two types were used: tRNA (Asp)4-ox and tRNA (Asp)15-ox). Formylase must be present at a concentration 10 times lower than that of tRNAox, i.e., at 10 µM in the 10 µL reaction mixture. This mixture also included1 µL of 50 mM sodium cyanoborohydride (NaBH3CN) and 1 µL of 10× buffer (a 10-fold more concentrated labeling buffer). The mixture was then incubated for 1 h at 37 °C. At the end of labeling, the mixtures were treated by adding 1.5 µL of freshly prepared 0.5 M sodium borohydride (NaBH4) in water (DNase/RNAase free) and 3.5 µL of 3× Laemmli; 15 µL portions of the NaBH4-treated samples were analyzed by electrophoresis on a 10% polyacrylamide gel under denaturing conditions (SDS-PAGE) revealed by Coomassie blue staining.

### 4.8. Statistical Analysis

Rootlet lengths by concentration were presented as mean values ± SE (standard error) for separate experiments using n seeds. Graphs of concentration–response curves were determined using nonlinear regression and were fitted to the Hill equation by an iterative least squares method (GraphPad Prism 8.0 Software, San Diego, CA, USA) to provide estimates of the maximum effective concentration IC_50_ (the negative logarithm of the agonist concentration producing 50% of maximum inhibition). For the comparison of the different effects against the control, one-way analysis of variance (ANOVA) was performed followed by multiple comparison t-tests. For a simple comparison of unpaired IC_50_, statistical significance was determined using the Bonferroni–Dunn method, with alpha = 0.05. Each row was analyzed individually, without assuming a consistent SD.

## 5. Conclusions

In conclusion, this study explored the anticancer potential of bioactive peptides derived from the enzymatic hydrolysis of bovine and human hemoglobin. The results obtained using two distinct approaches highlighted their promising potential as anticancer agents. The investigation of key parameters revealed the impact of various factors on the antimitotic activity of the peptides:

The results clearly demonstrated that the degree of hydrolysis (DH) exerted significant influence on the efficacy of these peptides. More specifically, DH levels, particularly those at3–10%, proved considerably more effective, causing a marked inhibition of rootlet growth, surpassing by three to four times those observed with a DH of 0, corresponding to the native form of hemoglobin. In addition, certain peptide fractions of bovine hemoglobin showed greater activity than those of human hemoglobin.

The α137–141 (NKT) peptide, the focus of our study, stood out by displaying exceptionally low IC_50_ values: only IC_50_ = 29 ± 1 µg/mL for bovine hemoglobin and IC_50_ = 48 ± 2 µg/mL for human hemoglobin. These figures were 10 to 15 times higher than those for other fractions, highlighting the extraordinary potential of this peptide in reducing cancer cell proliferation.

The structural characteristics of peptides played a crucial role in the biological effects observed. For example, peptide α137–141, characterized by its random coil structure and molecular weight of 654 Da, was compared with the antimicrobial peptide α1–32, derived from the enzymatic hydrolysis of bovine and human hemoglobin, which had an alpha helix structure and molecular weight of 3327 Da. The α137–141 peptide had an exceptionally low IC_50_ of only 0.053 mg/mL (53 ± 9 µg/mL), whereas the α1–32 peptide had an IC_50_ of 0.28 mg/mL (280 ± 31 µg/mL), which was ten times higher. Despite this discrepancy, α1–32 peptide retained significant inhibitory capacity. These observations were consistent with the antimicrobial activities previously observed for these two peptides, where the α137–141 peptide demonstrated up to 54-fold lower minimum inhibitory concentrations (MICs) against a specific strain, underlining its superior efficacy in combating pathogenic micro-organisms.

The in vitro results obtained using the second approach confirmed previous findings and reinforced the hypothesis that the inhibition of protein synthesis plays a crucial role in the anticancer mechanism of these peptides. Furthermore, these data suggest that samples containing α137–141 peptide (NKT) and total hydrolysates can modulate the interaction between FMTS and oxidized tRNA, opening up the possibility of influencing molecular binding mechanisms and creating a competitive situation that may compromise the ability of substrate tRNA to efficiently bind to ribosomal protein in their presence.

In sum, this study broadens the horizon for future research in the field of anticancer molecules. The specific properties of the α137–141 peptide, combined with its antimicrobial and antioxidant potential, make it an attractive candidate for the development of innovative treatments. However, to translate these discoveries into tangible clinical applications, further investigations and in vivo studies will be essential to confirm their efficacy and safety. These scientific advances are inspiring new hope in the fight against cancer, paving the way for effective targeted therapies.

## Figures and Tables

**Figure 1 ijms-24-15383-f001:**
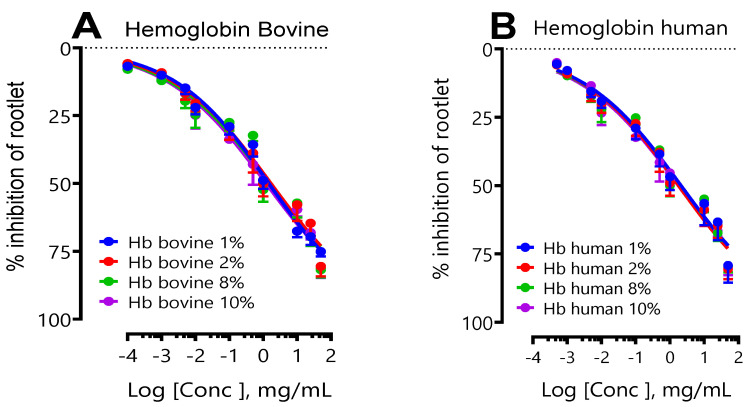
Effect of initial bovine (**A**) and human (**B**) hemoglobin concentration on LS rootlet growth. (**A**) Shows the effect of initial human hemoglobin concentration (1, 2, 8, and 10%) and (**B**) the effect of initial bovine hemoglobin concentration (1, 2, 8, and 10%).

**Figure 2 ijms-24-15383-f002:**
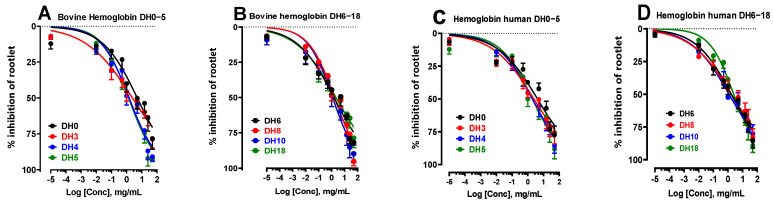
Effect of the degree of hydrolysis (DH) of bovine (**A**,**B**) and human (**C**,**D**) hemoglobin, i.e., a DH ranging from 0 to 18%, on the inhibition of LS rootlet growth.

**Figure 3 ijms-24-15383-f003:**
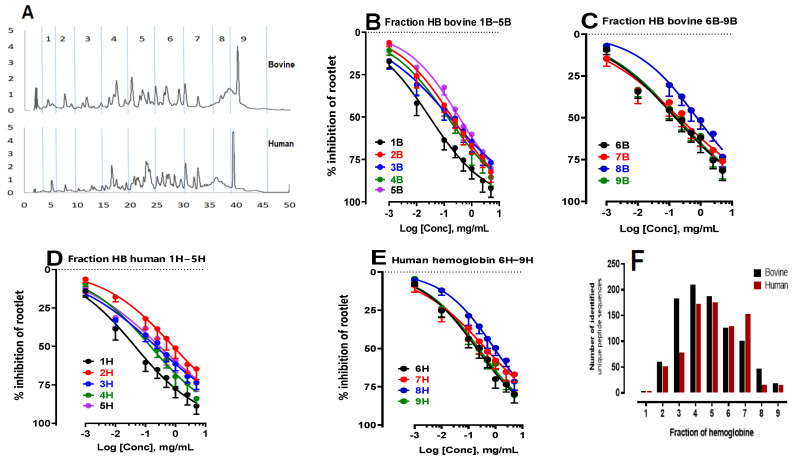
Study of the antimitotic activity and peptidomics analysis of human and bovine hemoglobin hydrolysate fractions after a 3h hydrolysis period (pH 3.5, 23°C, E/S = 1/11, CBH = 10%, CHH= 10%, *w*/*v*). Fractions were collected every 5 min. (**A**) Chromatographic profiles of bovine and human hemoglobin hydrolysis acquired with Empower 3 software (Version 3 Waters) at 215 nm by semipreparative HPLC, analyzed by a semipreparative C4 column. Effect of the influence of bovine (**B**,**C**) and human (**D**,**E**) hemoglobin hydrolysate fractions on the inhibition of LS rootlet growth. (**F**) Peptidomics analysis by UPLC-MS/MS and bioinformatics. Histogram showing the number of unique peptide sequences identified.

**Figure 4 ijms-24-15383-f004:**
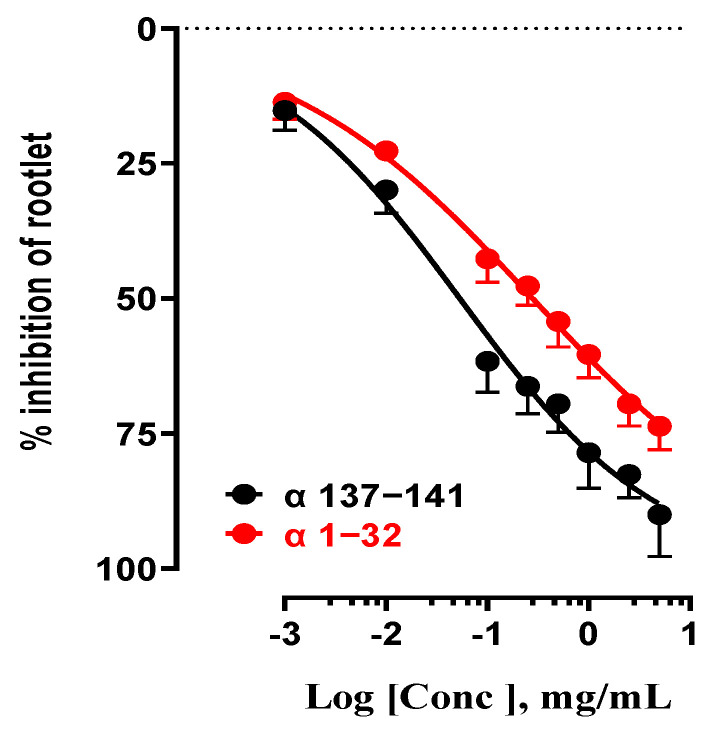
Effect of the influence of standard peptides α137−141 and α1−32 on LS rootlet growth inhibition.

**Figure 5 ijms-24-15383-f005:**
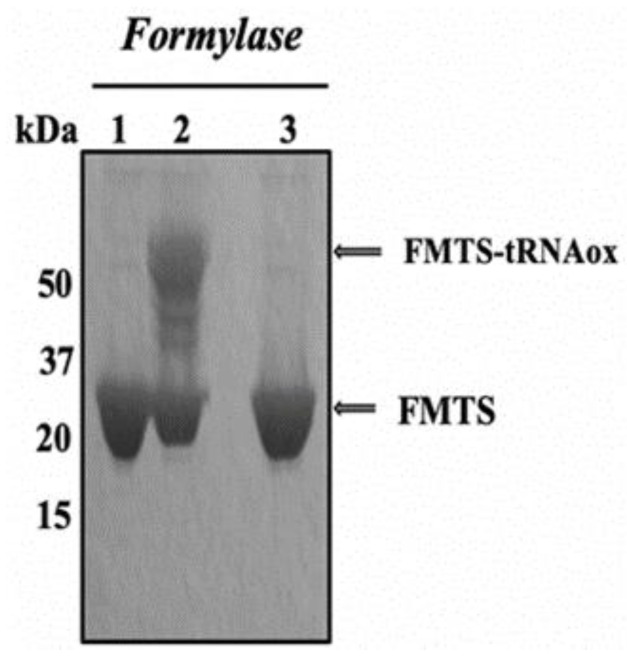
Covalent labeling of formylaseby tRNAox in the absence and presence of UCK-36. In well 1: free FMTS (formylase/3 µg/µL); in well 2: FMTS and tRNAox (36.5 µg/µL); in well 3: FMTS and tRNAox in the presence of 4-phenyl-3-thiosemicarbasone piperitone (UCK-36/1 mM of the UCK-36 molecule).

**Figure 6 ijms-24-15383-f006:**
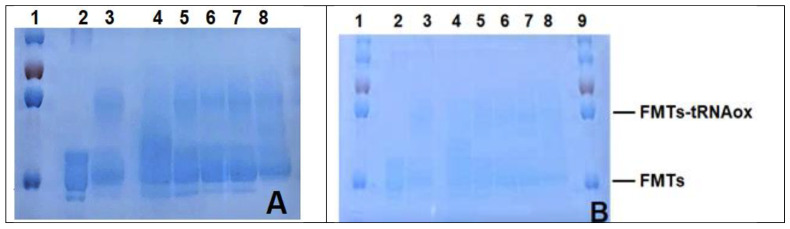
Testing of bovine (**A**) and human (**B**) hemoglobin peptide hydrolysates and their peptide fractions. Analysis was carried out by electrophoresis on a 10% polyacrylamide gel under denaturing conditions (SDS-PAGE), revealed by Coomassie blue staining. The targeted protein is FMTs. 1: marker (8 µL); 2: FMTs (0.3 µL); 3: FMTs (0.3 µL) + 6 µL of tRNAox; 4: FMTs (0.3 µL) + 6 µL of tRNAox + fraction 1 (NKT) (0.5 µL of 1/100); 5: FMTs (0.3 µL) + 6 µL of tRNAox + 2 (0.5 µL of 1/100); 6: FMTs (0.3 µL) + 6 µL of tRNAox + 3 (0.5 µL of 1/100); 7: FMTs (0.3 µL) + 6 µL of tRNAox + 4 (0.5 µL of 1/100); 8: FMTs (0.3 µL) + 6 µL of tRNAox + total hydrolysate (0.5 µL of 1/100); 9: marker (8 µL).

**Figure 7 ijms-24-15383-f007:**
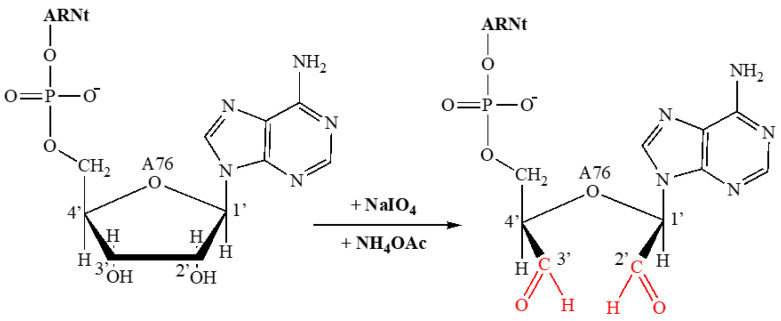
tRNA oxidation reaction.

**Figure 8 ijms-24-15383-f008:**
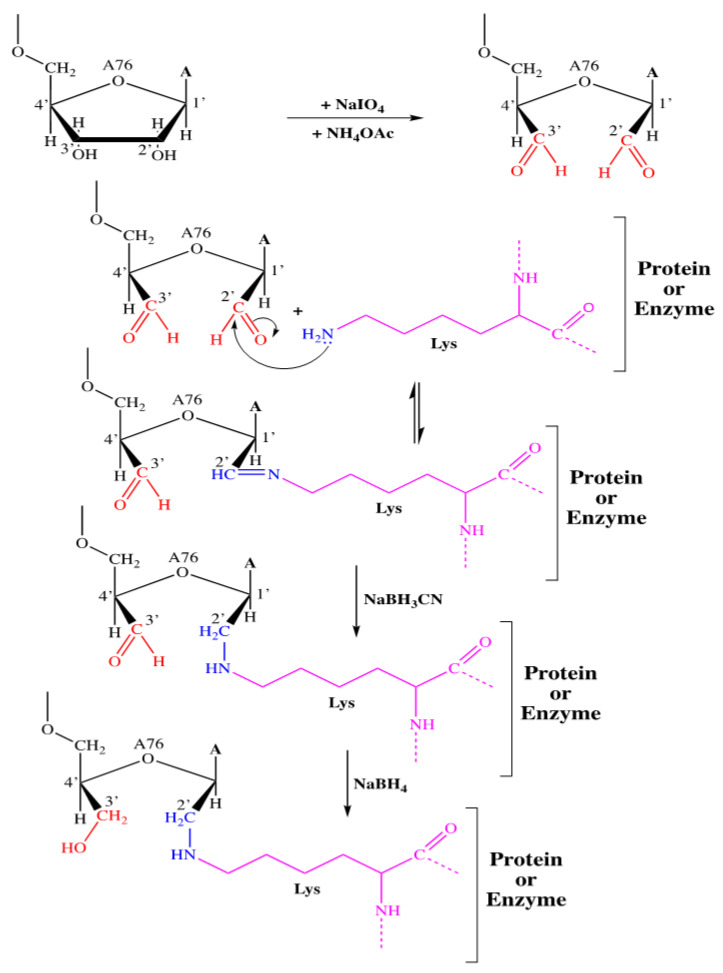
Global tRNAox-Lys (Protein)labeling reaction.

**Table 1 ijms-24-15383-t001:** Represents the IC_50_ values of bovine and human hemoglobin hydrolysates according to the initial hemoglobin concentration (1, 2, 8, and 10%).

	IC_50_ (mg/mL)	
(HB)	Bovine (B)	Human (H)
1	1.47 ± 0.65 ^ns^	2.12 ± 0.86 ^ns^
2	1.69 ± 0.70 ^ns^	1.80 ± 0.74 ^ns^
8	1.53 ± 0.61 ^ns^	2.24 ± 0.80 ^ns^
10	1.23 ± 0.52 ^ns^	1.77 ± 0.74 ^ns^

The IC_50_ values of the hydrolysates were compared with each other. ^ns^ indicates no significant difference. Statistical significance was determined using the Bonferroni–Dunn method, with alpha = 0.05. Each row was analyzed individually.

**Table 2 ijms-24-15383-t002:** IC_50_ values of bovine and human hemoglobin hydrolysates according to their DH.

Hydrolysis	IC_50_ (mg/mL)	
DH	Bovine (B)	Human (H)
DH0	5.06 ± 2.00 ^ns^	3.97 ± 0.60 ^ns^
DH3	2.42 ± 0.96 ^ns^	2.68 ± 0.67 ^ns^
DH4	1.46 ± 0.59 ^ns^	1.94 ± 0.47 ^ns^
DH5	1.35 ± 0.54 ^ns^	1.97 ± 0.84 ^ns^
DH6	1.54 ± 0.60 ^ns^	1.68 ± 0.67 ^ns^
DH8	1.68 ± 0.67 ^ns^	1.67 ± 0.65 ^ns^
**DH10**	**1.19 ± 0.47 ^ns^**	**1.17 ± 0.46 ^ns^**
DH18	2.25 ± 0.84 ^ns^	2.46 ± 1.06 ^ns^

The IC_50_ of the hydrolysates were compared to each other. DH10, in bold, was more effective than other DHs. ^ns^ indicates no significant difference. Statistical significance was determined using the Bonferroni–Dunn method, with alpha = 0.05. Each row was analyzed individually.

**Table 3 ijms-24-15383-t003:** Comparison of IC_50_ values of bovine and human hemoglobin hydrolysates according to their DH.

Comparison of IC_50_ (mg/mL)			
	Bovine (B)		Human (H)
1B	0.029 ± 0.001 *	1H	0.045 ± 0.002 *
2B	0.19 ± 0.08 *	2H	0.84 ± 0.35 *
3B	0.15 ± 0.06 ^ns^	3H	0.22 ± 0.09 ^ns^
4B	0.13 ± 0.05 ^ns^	4H	0.15 ± 0.06 ^ns^
5B	0.34 ± 0.14 ^ns^	5H	0.29 ± 0.11 ^ns^
6B	0.20 ± 0.08 ^ns^	6H	0.20 ± 0.08 ^ns^
7B	0.43 ± 0.17 ^ns^	7H	0.48 ± 0.19 ^ns^
8B	0.92 ± 0.40 ^ns^	8H	0.92 ± 0.40 ^ns^
9B	0.23 ± 0.10 ^ns^	9H	0.22 ± 0.09 ^ns^

The IC_50_ of the hydrolysates were compared to each other. ^ns^ indicates no significant difference. Values marked with a symbol (*) were significantly different. Statistical significance was determined using the Bonferroni–Dunn method, with alpha = 0.05. Each row was analyzed individually.

**Table 4 ijms-24-15383-t004:** The IC_50_ influence of standard peptides α137–141 and α1–32 on LS rootlet growth.

Bioactive Peptides	IC_50_ (µg/mL)
α137−141	53 ± 9 *
α1−32	280 ± 31 ^ns^

The IC_50_s of the antimicrobial peptides were compared to each other. Values marked with a symbol (*) where significantly differentness, (ns) indicates no significant difference. Statistical significance was determined using the Bonferroni–Dunn method, with alpha = 0.05. Each row was analyzed individually.

**Table 5 ijms-24-15383-t005:** Minimum inhibitory concentrations (MICs) of pure peptides from the hydrolysis of bovine hemoglobin isolated by HPLC-IP [33,36,37].

	Strains	Gram − *Escherichia. coli Salmonella. enteritidis*		Gram + *Staphylococcus. aureus Listeria. innocua Micrococcus. luteus*		
Peptides						
α1–32		154	54	38	38	90
α137–141		9	4.6	1	1	9

MIC expressed in μM.

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
