# Peer review of "Obtaining New Candidate Peptides for Biological Anticancer Drugs from Enzymatic Hydrolysis of Human and Bovine Hemoglobin"

_ijms, 2023, doi:10.3390/ijms242015383_

Round 1

Reviewer 1 Report

Outman et al. delved into the enzymatic hydrolysis of bovine and human hemoglobin, scrutinizing the antimicrobial attributes of the resulting hydrolysates. While the study didn't forge new ground in the realm of antimicrobial peptides, it presented a methodical workflow concerning the enzymatic hydrolysis of proteins, paving a way for further exploration into the mechanisms of protein synthesis. However, several specific areas necessitate further refinement as highlighted below:

1. The opening paragraph in the introduction seems rather extensive. It might enhance readability if it were broken down into several smaller paragraphs. Additionally, there appears to be a typographical error on line 65 where "MPAs" is mentioned.

2. Before delving into the nuances of hemoglobin hydrolysis on line 66, a brief overview of the prevalent methods to screen antimicrobial peptides could be beneficial. This might encompass discussions on chemical synthesis, insights from natural venom peptides, and approaches to enzymatic hydrolysis, among others.

3. For a more straightforward interpretation of the tables, it is suggested to replace the letter “a” with “n.s.” to indicate non-significance, and the letter “b” with “*” to mark significant findings.

4. In both Figure 4 and Table 2, it is unclear whether “DH 4,5” is a typographical error and was intended to be “DH5”. 

5. Within Table 2, is it possible to elucidate that certain hydrolysis conditions (particularly DH10) exhibit superior efficacy when compared to others?

6. The text mentions that fraction 1 was identified as NKT peptide. To bolster this assertion, it would be valuable to include the Mass Spectrometry (MS) spectra of Fraction 1 in Figure 5.

7. It would enhance Figure 5A if markers or indications were incorporated to pinpoint the locations of the original HB and NKT peptide.

8. A discrepancy seems to exist with respect to the labels in Table 3, where the values 2B and 2H are significantly different yet both are marked with “a”.

9. The manuscript would benefit from an extended discussion concerning the a1-32 peptide in section 2.1.4. Queries that might be addressed include: Has this peptide’s antimicrobial activity been explored in prior studies? Is it feasible for the enzymatic hydrolysis process to generate this peptide? If so, in which fraction would it be eluted?

10. It appears that the standard deviation of IC50 has been omitted in Table 4.

11. To aid readers in comprehending Figure 7, incorporating descriptions pertaining to the wells in the figure legend would be constructive.

Author Response

Reviewer 1

Comments and Suggestions for Authors

Outman et al. delved into the enzymatic hydrolysis of bovine and human hemoglobin, scrutinizing the antimicrobial attributes of the resulting hydrolysates. While the study didn't forge new ground in the realm of antimicrobial peptides, it presented a methodical workflow concerning the enzymatic hydrolysis of proteins, paving a way for further exploration into the mechanisms of protein synthesis. However, several specific areas necessitate further refinement as highlighted below:

  1. The opening paragraph in the introduction seems rather extensive. It might enhance readability if it were broken down into several smaller paragraphs. Additionally, there appears to be a typographical error on line 65 where "MPAs" is mentioned.

Response:  Corrections have been made

  1. Before delving into the nuances of hemoglobin hydrolysis on line 66, a brief overview of the prevalent methods to screen antimicrobial peptides could be beneficial. This might encompass discussions on chemical synthesis, insights from natural venom peptides, and approaches to enzymatic hydrolysis, among others.

Response: A paragraph has been added to the introduction in response to the reviewer's request

  1. For a more straightforward interpretation of the tables, it is suggested to replace the letter “a” with “n.s.” to indicate non-significance, and the letter “b” with “*” to mark significant findings.

Response:  Corrections have been made

  1. In both Figure 4 and Table 2, it is unclear whether “DH 4,5” is a typographical error and was intended to be “DH5”. 

Response: Corrections have been made

  1. Within Table 2, is it possible to elucidate that certain hydrolysis conditions (particularly DH10) exhibit superior efficacy when compared to others?

Response: Corrections have been made.

  1. The text mentions that fraction 1 was identified as NKT peptide. To bolster this assertion, it would be valuable to include the Mass Spectrometry (MS) spectra of Fraction 1 in Figure 5.

Response: There is no mass spectrometry (MS) data for Fraction 1 in Figure 5, but we conducted identification and quantification in the preceding article (Outman et al., 2023) published in your journal titled: "Outman, A.; Deracinois, B.; Flahaut, C.; Diab, MA; Gressier, B.; Eto, B.; Nedjar, N. Potential of human hemoglobin as a source of bioactive peptides: a comparative study of enzymatic hydrolysis with bovine hemoglobin and the production of active peptide α137–141. Int. J. Mol. Sci. 2023, 24, 11921. https://doi.org/10.3390/ijms241511921."

  1. It would enhance Figure 5A if markers or indications were incorporated to pinpoint the locations of the original HB and NKT peptide.

Response:  Corrections have been made

  1. A discrepancy seems to exist with respect to the labels in Table 3, where the values 2B and 2H are significantly different yet both are marked with “a”.

Response:  Corrections have been made

  1. The manuscript would benefit from an extended discussion concerning the a1-32 peptide in section 2.1.4. Queries that might be addressed include: Has this peptide’s antimicrobial activity been explored in prior studies? Is it feasible for the enzymatic hydrolysis process to generate this peptide? If so, in which fraction would it be eluted?

Response: A paragraph has been added to the discussion in response to the reviewer's request.

Yes, the α1-32 peptide is also derived from the enzymatic hydrolysis of both bovine (Nedjar-Arroume et al., 2008) and human hemoglobin (Mak et al., 2004). This α1-32 fragment exhibits significant antimicrobial activity against various bacterial strains, including Gram-positive bacteria such as Staphylococcus aureus, Listeria innocua, and Micrococcus luteus, as well as Gram-negative bacteria like Escherichia coli and Salmonella enteritidis (Catiau et al., 2011b; Nedjar-Arroume et al., 2006). In this study, a comparison was made with two synthetic peptides, and it is also identified in fraction 6 when using a C4 column, eluting between 25 and 30 minutes (Véronique Dubois in 2006 and Nedjar-Arroume in 2008).

  1. It appears that the standard deviation of IC50 has been omitted in Table 4.

Response: Corrections have been made

  1. To aid readers in comprehending Figure 7, incorporating descriptions pertaining to the wells in the figure legend would be constructive.

Response:  Corrections have been made

Reviewer 2 Report

ijms-2619367

Obtaining new candidate peptides for biological anticancer drugs from enzymatic hydrolysis of human and bovine Hemoglobin

1. Summary

The manuscript aims to obtain bioactive peptides by enzymatic hydrolysis of bovine and human hemoglobin and evaluate their effects on cell growth inhibition, therefore they may serve as potential candidates for biological anticancer drugs.

The study was performed by rapid screening of candidate anticancer peptides derived from enzymatic hydrolysis of bovine and human hemoglobin using the Lepidium sativum root growth technique.

Based on the obtained results, the present study shows that: 1) the degree of hydrolysis (DH) significantly affects the production of bioactive peptides. DH levels of 3-10% yield resulted in a significantly stronger inhibition of root growth compared to DH 0 (native form of hemoglobin), exhibiting three to four times greater intensity; 2) Certain peptide fractions of bovine hemoglobin show higher activity than that of human hemoglobin; 3) The structural characteristics of the peptides play a decisive role in the observed biological effects; 4) α137-141 peptide is the most active among fractions derived from bovine (IC50 = 29 ± 1 µg/mL) and human hemoglobin (IC50 = 48 ± 2 µg/mL), which stands out as 10 to 15 times more high power relative to other hemoglobin.

In the first part "Introduction", the authors clarify the role of antimicrobial peptides (AMPs) obtained by hydrolysis of hemoglobin, which represent a promising and underutilized alternative for the treatment of cancer compared to other methods used so far such as immunotherapy, horotherapy with drugs, transplantation of stem cells, biomarker testing and radiotherapy, chemotherapy and more. The advantages of biologically active peptides, which have a lower tendency to develop resistance compared to conventional chemotherapy treatments, have also been clarified. The role of the above-mentioned antimicrobial peptides is explained, adding their ability to protect against infections and also their strong toxicity towards cancer cells.

An analysis of the anticancer activity of bovine and human hemoglobin hydrolysates and their peptide fractions was performed using two different approaches: The first approach consists of using a screening process for drug candidates that inhibit protein synthesis and can be used as antibacterial and anticancer agents. The second approach focused on the in vitro anticancer activity of peptide hydrolysates of human and bovine hemoglobin, focusing on the eL42 protein as a potential target for the destruction of cancer cells.

The possibility of comparative analysis of the peptide populations in the peptide hydrolyzate fractions from hemoglobin by UPLC-MS/MS is also indicated.

In the "Results" section, the results obtained from the study of the effects of bovine and human hemoglobin hydrolysates and their peptide fractions on the inhibition of protein biosynthesis in cancer cells are presented and compared.

The study of the influence of the initial concentration of bovine and human hemoglobin on protein biosynthesis is described. Analysis of the IC50 for bovine and human hemoglobin showed generally equivalent values, indicating that varying the initial concentration (1%, 2%, 8%, 116 or 10%) did not result in a significant difference in the effect on inhibition of LS cell growth.

The growth inhibitory activity of bovine and human hemoglobin hydrolyzate roots was also investigated according to their degree of hydrolysis.  Due to necessity for the study are selected eight hydrolysates with DHs of 0, 3, 4, 5, 6, 8, 10, and 18%, corresponding to hydrolysis times of 0, 5, 15, and 30 min, as well as 1, 2, 3 and 24 hours respectively.

For some DH, bovine and human hemoglobin hydrolysates showed a more marked effect in inhibiting root growth than for others.

The other DH showed very interesting inhibitory effects on root growth, three to four times more intense than for DH of 0. For example, DH of 4, 5, 6, 8 and especially 10% showed significantly greater inhibitory effects.

The investigation of root growth inhibition by bovine and human hemoglobin hydrolyzate fractions using a peptidomic approach is also explained.

Fractionation of these hydrolysates was performed after a 3-hour peptic hydrolysis. The first fraction (fraction 1) corresponds to the NKT peptide (α137-141, neokyotorphin), followed by fractions collected at 5-min intervals.

Analysis of IC50 values revealed several fractions in both hemoglobin species that stood out for their strong inhibitory activity. Of the analyzed fractions, fraction 1 stands out for its remarkable inhibitory effect. The IC50 values for this fraction were 29 ± 1 µg/mL for bovine hemoglobin and 45 ± 2 µg/mL for human hemoglobin. Factors 3, 4, 5, 6 and 9 stand out for their significant inhibitory capacity against root growth.

By cross-comparison of the results, it was found that fractions with LS rootlet inhibitory properties in both species, especially fractions 3, 4, 5, 6 and 7, had the highest number of identified peptide sequences.

The study of the inhibition of the root growth activity of the pure antimicrobial peptides α137-141 and α1-32 with different structural characteristics is also described.

Also present is a detailed description of the assay of inhibition of root growth activity of bovine and human hemoglobin hydrolysates and their fractions as a consequence: Covalent labeling of Formylase by tRNAox and testing of the resulting peptide hydrolysates.

In the Discussion section they are described the advantages of using antimicrobial peptides (AMPs) obtained by hydrolysis of human and bovine hemoglobin. The issue of the influence of the initial concentration of hemoglobin was discussed, emphasizing that it does not particularly affect the growth of roots in both types of hemoglobin. Studies on the importance of the degree of hemoglobin hydrolysis on the inhibitory effects of the obtained hydpolysates are also commented. In particular, at a degree of hydrolysis, DHs of 3, 4, 5, 6, 8, and especially 10% showed significantly higher inhibitory effects than those observed at a DH of 0, and these effects were three to four times more intense. Despite the different peptide profiles from the two hemoglobin sources, fractions 3, 4, 5, 6 and 9 showed comparable inhibitory effects between cattle and humans. It is also emphasized that the hydrolysates and their fractions, in particular the NKT fraction, specifically target the protein synthesis mechanism of the host cell, a crucial aspect of limiting tumor proliferation. It is also indicated that the synergistic approach of targeting both HSP 90/70 and the 26S proteasome emerges as a promising strategy for cancer treatment, highlighting the crucial role of multifactorial approaches. The advantages of the method of inhibiting the growth activity of the roots of bovine and human hemoglobin hydrolysates and their fractions are also described: Covalent labeling of Formylase by tRNAox and testing of the resulting peptide hydrolysates in cancer treatment compared to other methods. It is also emphasized that the synergistic approach of targeting both HSP 90/70 and the 26S proteasome emerges as a promising strategy for cancer treatment, highlighting the crucial role of multifactorial approaches. The results of this study suggest that samples containing the 1(NKT) fraction and the total hydrolyzate for both types of hemoglobin may exert modulatory effects on the interaction between HTSF and tRNAox. The observations made in the study suggest that NKT may be one of the most promising pathways for inhibitory activity. In addition, its significant antimicrobial and antioxidant properties may qualify it as a potential candidate for inclusion in the class of anticancer agents. However, further investigations are indicated to be extremely important to detail the mechanisms underlying these inhibitory effects and to better understand the impact of our bioactive peptides on the potential competition with transfer RNA (tRNA) for binding to the eL42 protein in the human 80S ribosome.

The Materials and Methods section describes in detail all reagents, solvents and standards used to carry out the study; the method of preparation of hydrolysates from bovine and human hemoglobin; preparation of the basic solutions of hemoglobin, the hydrolysis of the two types of hemoglobin; the fractionation of peptide hydrolysates by semi-preparative HPLC; RP-UPLC analysis and mass spectrometry; the determination of anticancer activity of the hydrolysates and peptide fractions and the statistical processing of the data by variance analysis and non-linear regression.

2. Overall opinion

The manuscript is of potential interest to scientists with interests in medicine and biochemistry, physicians, pharmacists and decision makers, but needs some revision before publication to ensure better structure and flow. The text should be revised and organized.

The English format needs revision regarding word forms and correction of grammatical and typographical errors in the text.

3. Comments

The summary is too short. No other methods of fighting cancer and numerical data from the study are indicated.

The Introduction section is well structured. Methods for fighting cancer and the advantages of using antimicrobial peptides obtained by the hydrolysis of human and bovine hemoglobin to inhibit protein synthesis in cancer cells and stop their development are discussed.

In the Discussion section, it is necessary to indicate more authors who have done similar studies to be compared with the present study.

All graphics placed in the text must be enlarged.

Lines 677 (number 24) and 691 (number 30) do not indicate the issue number of the magazine and the pages on which the information is found.

The conclusion contains no numerical data to support the findings stated in the discussion. It is well structured and recommendations for future work are given.

The English format needs revision regarding word forms and correction of grammatical and typographical errors in the text.

Author Response

  1. Overall opinion

The manuscript is of potential interest to scientists with interests in medicine and biochemistry, physicians, pharmacists and decision makers, but needs some revision before publication to ensure better structure and flow. The text should be revised and organized.

The English format needs revision regarding word forms and correction of grammatical and typographical errors in the text.

  1. Comments

The summary is too short. No other methods of fighting cancer and numerical data from the study are indicated.

Response:  Corrections have been made

The Introduction section is well structured. Methods for fighting cancer and the advantages of using antimicrobial peptides obtained by the hydrolysis of human and bovine hemoglobin to inhibit protein synthesis in cancer cells and stop their development are discussed.

Response:  Corrections have been made

In the Discussion section, it is necessary to indicate more authors who have done similar studies to be compared with the present study.

Response:  Corrections have been made

All graphics placed in the text must be enlarged.

Response:  Corrections have been made

Lines 677 (number 24) and 691 (number 30) do not indicate the issue number of the magazine and the pages on which the information is found.

Response: Reference number 24 corresponds to a patent, while document number 30 is a thesis manuscript.

The conclusion contains no numerical data to support the findings stated in the discussion. It is well structured and recommendations for future work are given.

Response:  Corrections have been made

Round 2

Reviewer 1 Report

In Table 4, the letters “a” and “b” need to be revised accordingly. Overall, this manuscript is in a good format and ready for publication.

Author Response

Corrections have been made in table 4